# The Influence of Research Follow-Up during COVID-19 Pandemic on Mental Distress and Resilience: A Multicenter Cohort Study of Treatment-Resistant Depression

**DOI:** 10.3390/ijerph19063738

**Published:** 2022-03-21

**Authors:** Pham Thi Thu Huong, Chia-Yi Wu, Ming-Been Lee, Wei-Chieh Hung, I-Ming Chen, Hsi-Chung Chen

**Affiliations:** 1School of Nursing, National Taiwan University College of Medicine, Taipei 10051, Taiwan; phamhuong@hmu.edu.vn (P.T.T.H.); evieh90@gmail.com (W.-C.H.); 2Faculty of Nursing and Midwifery, Hanoi Medical University, Hanoi 116177, Vietnam; 3National Institute of Mental Health, Bach Mai Hospital, Hanoi 116300, Vietnam; 4Department of Nursing, National Taiwan University Hospital, Taipei 10051, Taiwan; 5Taiwan Suicide Prevention Center, Taiwanese Society of Suicidology, Taipei 10051, Taiwan; mingbeen@ntu.edu.tw; 6Department of Psychiatry, Shin Kong Wu Ho-Su Memorial Hospital, Taipei 11101, Taiwan; 7Department of Psychiatry, National Taiwan University Hospital, Taipei 10051, Taiwan; b90401022@gmail.com (I.-M.C.); hsichungchen@ntu.edu.tw (H.-C.C.); 8Department of Nursing, Far Eastern Memorial Hospital, New Taipei City 220, Taiwan; 9Center of Sleep Disorders, National Taiwan University Hospital, Taipei 10051, Taiwan

**Keywords:** COVID-19, treatment-resistant depression, resilient coping, mental distress, follow-up study

## Abstract

Background: During the COVID-19 outbreak, patients with mental disorders have faced more negative psychological consequences than the public. For people with treatment-resistant depression (TRD), it is unclear whether research engagement would protect them from the deterioration of their symptoms. The study aimed to examine if chronic depressive patients would have improved resilience and mental distress levels after follow-up interviews during an observation period under COVID-19. Methods: The study was nested within a three-year prospective cohort study. A two-group comparison design was conducted, i.e., the follow-up group with regular research interviews every three months after baseline assessment and the control group with one assessment-only interview. The two groups were compared with demographics, psychosocial, and suicide information. Results: Baseline assessments were not significantly different in sociodemographic variables, suicide risks, mental distress, and resilience between groups. Significant differences were detected in resilient coping and mental distress levels (*p* < 0.05). The follow-up group (*n* = 46) experienced a higher level of resilient coping (37% vs. 25%) and lower level of mental distress (47.8% vs. 64.7%) than the control group (*n* = 68). Conclusions: Findings highlight under universal government strategy against COVID-19, TRD patients receiving regular research follow-ups exhibited better resilience and less mental distress than those without regular support from healthcare providers.

## 1. Introduction

The novel coronavirus disease 2019 (COVID-19) was first detected in Wuhan, China, in late 2019. Taiwan developed extraordinary government strategies in big data analytics, cutting-edge technological advances, and proactive testing [1]. Quick responses from the government, such as broadcasts of appropriate wearing masks, washing hands, mask purchase policy, and social distancing policy [1,2] with efficient risk control plan of actions in hospitals [3,4], have reassured and satisfied Taiwanese citizens during the crisis. The nationwide approaches recommended by the Taiwan Ministry of Health and Welfare, including the use of the App “The Mood Thermometer” (5-item Brief Symptom Rating Scale) to self-monitor mental distress and the restriction of exposure to COVID-19-related news, had resulted in positive impacts [5].

Given the challenges of public health mechanisms responding to COVID-19, the three most significant psychological stressors were identified, i.e., fear of infection, finance issues, and home quarantine [6]. As a result, psychological distress was a common symptom among the public [6,7,8,9] and the medical staff [10] during the COVID-19 outbreak. In addition, the pandemic has significantly impacted health behavior alongside mental health consequences [11]. Chinese adults (≥18 years old) reported a clinically elevated prevalence of anxiety and depression at the peak of the pandemic that was 5-fold higher than the prevalence in 2019 [6]. Young adults aged 18–30 years in the United States reported a sharp increase in depression and anxiety during COVID-19 compared to young adults in 2006 [8,12] or primary care patients in 2005 [13]. Most young adults reported low resilience (72.0%), low distress tolerance (74.1%), and loneliness (61.5%), which significantly increased the levels of depression, anxiety, and post-traumatic stress disorder [8]. In this context, it is universally acknowledged that both coping and resilience levels of those who can manage an emotional response and overcome mass trauma would grow more robust than before [14]. Resilience is defined as the ability to rise above difficult situations [15] and innate human capacity in dealing with tremendous challenges [16]. The literature points out that resilience is a protective factor against the development of mental disorders such as depression and suicide or in minimizing the severity of illness [17]. A cross-sectional study has addressed that resilience plays moderating and mediating roles in the associations of the personality traits with depressive symptoms [18], and resilience is represented as an essential target intervention in public health emergencies during the COVID-19 epidemic [19]. However, few previous studies applied longitudinal designs to study the trend of resilience among patients with depression. It is also unclear whether the psychological distress level fluctuates and whether resilience exists among the middle- and old-aged adults with depression during the COVID-19 pandemic.

Patients with mental disorders are categorized as the most vulnerable group in COVID-19 that needs further intervention, high-risk identification, and follow-ups [20,21,22], particularly those with long-term morbidity such as treatment-resistant depression (TRD). TRD is commonly defined as “a minimum of two prior treatment failures and confirmation of prior adequate dose and duration for patients with the major depressive disorder” [23]. It was found to be prevalent in one-third of patients with depression [24] and is characterized by functional impairment, lower quality of life, productivity loss, comorbidities, higher medical costs [25], and increased risk of mortality or suicide ideation [25,26,27] compared with non-TRD or the general population. Thus, patients who suffer from TRD tend to be chronic with partial remissions in the long-term recovery process. Due to the reduction of cognitive functions, patients with severe mental illness face many difficulties in following the infection control instructions [15,16]. Hence, these patients were categorized as the most vulnerable group in COVID-19 that required high-risk identification, intervention, and follow-ups [20,21,22]. Moreover, low levels of resilience and high distress tolerance were significantly associated with depression [8,28]. It is crucial to examine the role of resilience and distress under the threat of COVID-19 outbreak in TRD patients to provide timely intervention.

Since the pandemic′s prospective psychological impact remains unclear among patients with chronic mental illnesses [20,29], the study hypothesized that research follow-ups through the outbreak would enhance the psychological conditions of patients with TRD due to the provision of perceived social support from the research personnel. It was based on the theory that “perceived social support” would affect depressive symptoms to a more considerable extent than received support [30,31] during the COVID-19 pandemic. The research question was whether follow-up interviews with perceived support would be related to positive psychological changes such as resilience and mental distress among the TRD patients.

## 2. Materials and Methods

### 2.1. Study Design and Participants

This was a two-group observational study nested within a three-year prospective cohort study project, which was ethically approved by two ethics committees of two study hospitals in northern Taiwan (ID: 201612198RINB and 20190106R). During the COVID-19 outbreak in 2020, the research team observed a TRD cohort in terms of the levels of resilience and psychological distress before and during the pandemic. Two telephone interviews were arranged at two-time points (baseline and second interviews during January and May 2020), comparing the conditions of psychological resilience and distress between the two groups (i.e., the follow-up group and the control group) of the cohort. The TRD cohort included 125 patients with treatment-refractory depression referred by two psychiatrists in the study hospitals. The follow-up group consisted of 46 patients who received trimonthly regular follow-up using a predesigned questionnaire; other patients from the cohort (*n* = 68) (i.e., the control group) were not regularly contacted until the follow-up interview during the COVID-19 outbreak. In total, there were 11 patients lost to follow-up or with missing values that were excluded from the study (attrition rate: 8.8%).

### 2.2. Measures

#### 2.2.1. Demographic Information

Personal demographics were collected by variables including age, education years, gender, marital status (single/married or cohabited/divorced or separate/widow), religious belief (yes/no), and employment status (yes/no).

#### 2.2.2. The Five-Item Brief Symptom Rating Scale (BSRS-5)

The 5-item Likert scale, also named the “Mood Thermometer”, assesses the psychological distress level in the past week by self-report. It contains the following five items of psychopathological symptoms: (1) having any sleep problems (insomnia); (2) feeling tense or keyed up (anxiety); (3) feeling easily annoyed or irritated (hostility); (4) feeling low in mood (depression); (5) feeling inferior to others (inferiority). All items were scored by 0–4 points (0, not at all; 1: a little bit; 2, moderately; 3, quite a bit; 4, extremely). An additional question, “Do you have any suicide ideation?” was added at the end of the scale to assess the patient’s recent suicide ideation in the past week. A total score of BSRS-5 under 5 may indicate a low level of mental distress, a score between 6 and 9 could indicate moderate mental distress, and a score above 10 may indicate severe mental distress [32]. The Cronbach’s α for the BSRS-5 in this study was 0.84.

#### 2.2.3. The Nine-Item Concise Mental Health Checklist (CMHC-9)

The scale was designed for the rapid screening of psychological distress and the overall suicide risk levels of the participants. The nine items were divided into two core components for assessment, i.e., psychopathology and suicidality (major suicide risk factors). Each item was rated as yes/no and scored one point if present for the condition (e.g., stated future intent); the total score ranged from 0 to 9, with the cutoff score greater than 4 points indicating higher suicide risk compared to those with a score of 4 or less. A higher score indicates a higher level of the overall risk of suicidal behavior. The Cronbach′s alpha of the CMHC-9 was fair (α = 0.79) in this study, with a two-factor structure of psychopathology and suicidality by exploratory factor analysis [33].

#### 2.2.4. The Brief Resilient Coping Scale (BRCS)

The 4-item scale was designed to measure a person’s tendency to resiliently cope with stress, including the assessment of active coping with loss, positive growth, problem-solving, and self-control [34]. Each item uses a 5-point Likert scale from 1 = describes me not at all to 5 = describes me very well. The sum score ranges from 4 to 20 with scores of 4–10 indicating low resilient coping, 11–14 indicating medium resilient coping, and ≥15 indicating high resilient coping. The previous study indicated that the BRCS had satisfactory reliability estimated by Cronbach’s alpha (0.69) and test–retest reliability (0.71) among patients with rheumatoid arthritis. The BRCS was also reported to have significant associations with complimentary personal coping resources, adaptive pain-coping behaviors, and psychological well-being [34]. The Cronbach’s α for the BRCS in this study was 0.82.

### 2.3. Study Procedure

Under the COVID-19 period between January and May in 2020, the TRD subjects were followed up as scheduled in the study. A research assistant was trained by the principal investigator to communicate with the study participants through telephone interviewing. The interview contents and procedure were developed based on clinical experience, with two questions related to COVID-19 life changes included in the interview after accomplishing the structured measurements. These topical questions of interest were as follows: (1) How was your life affected under COVID-19? (2) How have you been coping with psychological symptoms since the last time we talked? With the standardized procedure, the research team assessed the subjects’ life changes and the outcomes through caring and empathetic attitudes and provided government anti-infection and self-care information to support them during the outbreak. Each participant was interviewed for around 30 min twice during the study period with a duration of three months between interviews. Baseline interviews at T0 used the same assessments as those used at T1. The participants’ responses were recorded by the assistant on a piece of paper and then keyed on the computer for statistics by the same person.

### 2.4. Statistical Analysis

The descriptive statistics were performed to analyze the participant profile (i.e., age, gender, education year, religion, employment, and marital status), suicide risks, mental distress, and resilience. The between-group comparison was analyzed either by independent *t*-tests or χ^2^ tests. Missing data were excluded. Further, the 11 participants lost to follow-up with missing values were due to death, low contact, and refusal to participate. Mental distress and resilience levels were graphed to show their fluctuations between two groups of TRD patients during COVID-19. The analyses were performed with the statistical software SPSS 22.

## 3. Results

### 3.1. The Participants’ Demographics

A total of 114 participants responded to the study from a previously established cohort of 125 patients who met the study criteria of TRD. Table 1 shows the distributions of the basic profile of the participants at baseline. There are 46 patients in the follow-up group and 68 patients in the control group. In total, the mean age of the two groups was 56.9 (SD = 14.4), with 11.9 (SD = 4.7) years of education. The number of female participants was 2.45-fold higher than that of males in this study and was allocated comparably in the two groups. The two groups were found to have no significant differences in marital status, religion, and unemployment status (Table 1).

### 3.2. Suicide Risk Factors of the Participants at Baseline Interview

Table 2 shows the suicide risk variables of the participants at the baseline interview. Nearly half of the participants (*n* = 48, 42%) disclosed suicide ideation one week before the interview. Specifically, the overwhelming majority of samples revealed suicide ideation during their life (*n* = 109, 95%). Over half (*n* = 66, 57.9%) had experienced at least one suicide attempt. About one-fifth of patients in the control group had a history of family member death from suicide (*n* = 14, 20.6%), a proportion which was slightly higher than that in the follow-up group (*n* = 7, 15.2%). While 16.2% (*n* = 11) of patients in the control group reported a history of the family attempted suicide, only 4.3% (*n* = 2) of patients in the follow-up group did. No significant differences were identified in the above results according to Fisher’s exact statistics. All patients in two groups reported recent suicide ideation, which was identified by the total score of the 9-item CMHC with a mean score of 4.04 ± 2.60 (Table 2).

### 3.3. Trend of Mental Distress and Resilience before and after the COVID-19 Outbreak

The results, given in Table 3, show mental distress and resilience before and during the COVID-19 outbreak. A significant reduction was detected in all mental distress levels in the follow-up group compared to the control group (*p* < 0.05). The percentage of patients with severe mental distress (BSRS-5 score ≥10) decreased from 67.4% (*n* = 31) to 47.8% (*n* = 22) in the follow-up group; in contrast, there was a minor decline in the control group from 67.6% (*n* = 46) to 64.7% (*n* = 44) (Table 3).

As Figure 1 shows, there were fluctuations in mental distress during COVID-19 between the two groups. Patients in the follow-up group made more significant improvements in overall BSRS-5 score, from 12.4 (SD = 6.6) to 9.8 (SD = 6.7), than those in the control group (M = 11.8 (SD = 5.9) to M = 11.1 (SD = 7.2)). However, the difference in the BSRS-5 total score was not significant.

In terms of the fluctuation in resilience between the two groups, both the total score and three levels of resilience increased significantly in the follow-up group compared to the control group (*p* < 0.05). At baseline, the follow-up group had a total BRCS score indistinguishable from that of the control group (M = 11.5 (SD = 3.9) vs. M = 11.8 (SD = 3.8)). However, during our observation period, the two groups’ scores were significantly different (follow-up M = 13.2 (SD = 3.3) vs. control M = 10.8 (SD = 4.2)) (*p* < 0.01). Such change can be illustrated more clearly in Figure 2. The inequity of resilience levels significantly appeared in tertiles (*p* < 0.05). There was a notable improvement of 1.88-fold in the number of patients who were in the high level of resilient coping (from 19.6% (*n* = 9) to 37% (*n* = 17)) in the follow-up group compared to a slight improvement from 22% (*n* = 15) to 25% (*n* = 17) in the control during our observation period.

## 4. Discussion

In this study, the participants, a group of the TRD cohort, were followed up during COVID-19 in 2020. We examined whether a supportive research engagement would make a difference in the levels of resilience and mental distress between the participants who were with or without a follow-up interview. The findings showed that the follow-up group of TRD patients experienced a higher level of resilience and a lower level of mental distress than the control group. The results were interpreted under government strategies against COVID-19 and research follow-ups that supported these patients resiliently coping with the pandemic threats in Taiwan.

Patients with mental disorders were at higher risks of relapse due to fear related to the COVID-19 outbreak [35]. Current research on resilience and mental distress under the COVID-19 pandemic has been scarce, with only a few cross-sectional studies concerning young adults [8] and healthcare workers [36]. However, resilience was a protective factor against stress during and after the pandemic [14,16]. A study in SARS survivors after 6, 12, and 18 months of follow-up identified that those with higher levels of resilience and recovery experienced less SARS-related anxiety and more significant social support [37]. The result is consistent with the current findings. Moreover, perceived social support was regarded as correlate with coping strategy and less depression or anxiety [38,39], indicating the critical role of support from others during the pandemic. In this study, the TRD patients in the follow-up group had better-coping behavior in response to the life situations during the pandemic than their counterparts without periodical follow-ups. This could be explained by the fact that regular interviews with standardized topic guides as communication media by well-trained healthcare providers may help TRD patients better manage their emotional responses to the threats or life changes brought by COVID-19. A possible reason for this improvement may remain due to the Hawthorne effect [40]. However, if patients suffering from TRD can be encouraged to engage with life during tremendous challenges, then it may not be considerably problematic whether the Hawthorne effect is the cause of increasing mental well-being. Similarly, being screened and having an awareness of the research monitoring can lead to a change in drinking behavior in university students [41]. This concept has a significant implication for follow-up interviews as an intervention for enhancing behavioral and psychological health.

Furthermore, unlike natural disasters such as earthquakes and floods in which people gather to support each other, COVID-19 requires the need for social distance and wearing masks. Hence, the fear of COVID-19 increases individuals’ depression, anxiety, and stress levels above the normal condition [42]. Therefore, in addition, to help to improve psychological support routinely to enhance resilience and psychoneuroimmunity against the epidemic [20], mental health providers and other allied professionals should educate the TRD patients and their families about how to cope with stress through appropriate contact with mental health services or information resources to reduce distress and anticipatory anxiety. A recommendation from the Taiwan Ministry of Health and Welfare suggested that COVID-19-related news contact for less than 30 min per day can be resourceful and protective to provide practical and safe support for the public [5]. The negative effect resulting from overexposure to COVID-19-related news should be noted and prevented. On the other hand, perceived support provided through social contact over virtual or actual visits may reduce depression or psychological distress under the pandemic [31]. Such mental support, when offered proactively through research follow-ups like those performed in the current study, may enhance the feelings of safety and perceived belongingness, thus maintaining psychological well-being.

This study showed that the majority of TRD participants revealed suicide ideation during their lifetime; over half had experienced suicide attempts, and nearly half reported suicide ideation within one week during the baseline interview. Our findings were consistent with other studies, which indicated that suicide attempt history and mental illness comorbidity were influential factors associated with suicide attempts for TRD patients [27]. During the time of COVID-19, remarkable interventions have been implemented to prevent human-to-human transmission. At the moment, these strategies seem to reduce the rate of infections. However, the potential for suicide risk is increasing [43], especially due to social distancing and its consequences in economic, psychological, and health-related risk factors [43,44]. This robust evidence highlights the need for comprehensive care, and long-term suicide prevention approaches for TRD patients during pandemic upheavals, such as regular free counseling and self-help through tele-mental health [22,43]. The finding suggested that under major stress caused by an environmental and health crisis, TRD patients may need both governmental precautions and periodical mental support from psychiatric and mental health professionals to promote psychological resilience and treat mental health conditions. Due to the uncertain future wave of COVID-19 disease transmission, the psychological consequences resulting from infection control or environmental changes need to be followed up for longer terms.

## 5. Limitations

The current study is limited by the sample size and its duration of follow-up observation. The study participants were recruited from a TRD cohort established by the research team in 2018, which may potentially have led to selection bias due to limited recruitment sources within two general hospitals. Such bias may limit the power of generalizability. However, the research team had made efforts in recruitment. Further, due to fluctuations in the severity of TRD and COVID-19 outbreak conditions, the patients were more difficult to contact and recruit. While the research team tried to make appointments, patients’ concerns about hospital safety interfered with the sample size in two hospitals. During COVID-19, most of the interviews were conducted via telephone by a well-trained research assistant due to patients’ concerns about virus transmission. However, we believe that the level of support perceived by the participants through empathetic listening and proactive screening by the research team over the phone should be sufficient to maintain the research relationship. Despite these limitations, the findings precisely measured psychological reactions and made comparisons before and during COVID-19 using an established cohort of TRD patients. Due to well-established research relationships with the cohort, the study team could maintain contact with most participants to complete the questionnaire. To collect comparable responses, the research team had designed a standardized form of questions to draw the answers and explore related life changes during the COVID-19 outbreak, so we believe the results were reliable, consistent, and comparable between the two groups.

## 6. Conclusions

The findings revealed that patients who suffered from TRD possessed resilient potentials under the threat of the COVID-19 pandemic. Under close surveillance of the pandemic in Taiwan, health professionals’ engagement with psychiatric patients would benefit the TRD patients in coping with the pandemic resiliently under mental health support. Due to the uncertain future waves of COVID-19 disease transmission, it may take months or years to draw a coherent picture of the long-term mental health impact of the coronavirus. For better psychological outcomes, mental health care personnel and government support must pay more attention to the needs of TRD patients in the community settings under COVID-19 to offer proactive support, promote resilience, and reduce distress. These findings further advocate the need for further research and interventions for those with chronic depression to develop long-term recovery and resilience against environmental and other sources of stress.

## Figures and Tables

**Figure 1 ijerph-19-03738-f001:**
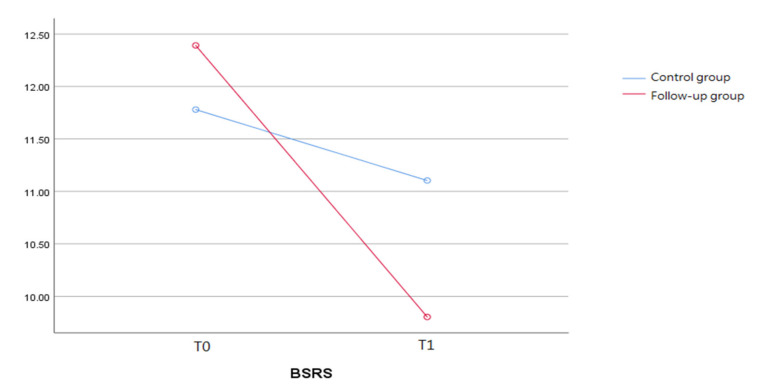
The fluctuations in mental distress across COVID-19 between the two groups.

**Figure 2 ijerph-19-03738-f002:**
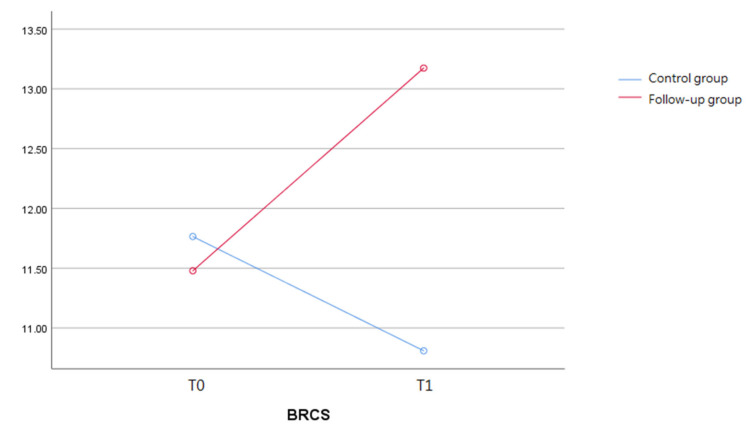
The fluctuations in resilience level across COVID-19 between the two groups.

**Table 1 ijerph-19-03738-t001:** Participant profile (N = 114).

*n* (%)/Mean ± SD	Total	FG (*n* = 46)	CG (*n* = 68)	X^2^/t *
Age (years)	56.9 ± 14.4	57.6 ± 12.0	56.4 ± 15.8	−0.42 (ns)
Educational years	11.9 ± 4.7	11.9 ± 4.3	11.9 ± 5.0	−0.01 (ns)
Gender				
Male	33 (28.9)	14 (30.4)	19 (27.9)	−0.29 (ns)
Female	81 (71.1)	32 (69.6)	49 (72.1)	
Marital status				
Single	22 (19.3)	9 (19.6)	13 (19.1)	1.49 (ns)
Married/cohabited	64 (56.1)	28 (60.9)	36 (52.9)	
Divorced/separate	18 (15.8)	5 (10.9)	13 (19.1)	
Widow	10 (8.8)	4 (8.6)	6 (8.8)	
Religion				
Yes	86 (75.4)	35 (76.1)	51 (75.0)	0.02 (ns)
No	28 (24.6)	11 (23.9)	17 (25.0)	
Employment status				
Yes	26 (22.8)	11 (23.9)	15 (22.1)	0.05 (ns)
No	88 (77.2)	35 (76.1)	53 (77.9)	

* Chi-square or t values with all nonsignificant (ns) findings; Fisher’s exact test applied in items with numbers less than 5; SD: standard deviation; FG: follow-up group; CG: control group.

**Table 2 ijerph-19-03738-t002:** Suicide risk factors of the participants at baseline interview.

*n* (%)/Mean ± SD	Total	FG (*n* = 46)	CG (*n* = 68)	X^2^/t *
Suicide ideation				
One Week	48 (42.0)	19 (41.3)	29 (42.6)	0.02 (ns)
Lifetime	109 (95.6)	44 (95.7)	65 (95.6)	−0.02 (ns)
Suicide attempt				0.28 (ns)
None	48 (42.1)	18 (39.1)	30 (44.1)	
Once	19 (16.7)	8 (17.4)	11 (16.2)	
Twice or more	47 (41.2)	20 (43.5)	27 (39.7)	
Family suicide history				4.87 (ns)
None	80 (70.2)	37 (80.4)	43 (63.2)	
Suicide attempt	13 (11.4)	2 (4.3)	11 (16.2)	
Death from suicide	21 (18.4)	7 (15.2)	14 (20.6)	
CMHC-9				
Item 1: Insomnia, past week	53 (46.5)	23 (50.0)	30 (44.1)	−0.61 (ns)
Item 2: Anxiety, past week	47 (41.2)	20 (43.5)	27 (39.7)	−0.40 (ns)
Item 3: Irritability, past week	54 (47.4)	21 (45.7)	33 (48.5)	0.30 (ns)
Item 4: Depressed mood, past week	62 (54.4)	23 (50.0)	39 (57.4)	0.77 (ns)
Item 5: Inferiority, past week	58 (50.9)	25 (54.3)	33 (48.5)	−0.61 (ns)
Item 6: Suicide attempt/self-harm, lifetime	68 (59.6)	30 (65.2)	38 (55.9)	−0.99 (ns)
Item 7: Alcohol/drug abuse, lifetime	42 (36.8)	14 (30.4)	28 (41.2)	1.16 (ns)
Item 8: Stated future suicide intent	21 (18.4)	6 (13.0)	15 (22.1)	1.22 (ns)
Item 9: Lack of social support	55 (48.2)	23 (50.0)	32 (47.1)	−0.31 (ns)
Total	4.04 ± 2.60	4.02 ± 2.85	4.04 ± 2.44	0.05 (ns)

* Chi-square or t values with all nonsignificant (ns) findings; Fisher’s exact test applied in items with numbers less than 5; SD: standard deviation; FG: follow-up group; CG: control group. CMHC-9: the 9-item Concise Mental Health Checklist.

**Table 3 ijerph-19-03738-t003:** The trend of mental distress and resilience before and during the COVID-19 outbreak (N = 114).

	T0 ^a^			T1		
*n* (%)/Mean ± SD	FG ^b^ (*n* = 46)	CG (*n* = 68)	X^2^/t	FG (*n* = 46)	CG (*n* = 68)	X^2^/t
*BSRS-5*						
Total scores	12.4 ± 6.6	11.8 ± 5.9	−0.52	9.8 ± 6.7	11.1 ± 7.2	0.97
Mental distress levels						
Low (0–5)	8 (17.4)	11 (16.2)	0.04	15 (32.6)	21 (30.9)	7.32 *
Moderate (6–9)	7 (15.2)	11 (16.2)		9 (19.6)	3 (4.4)	
Severe (≥10)	31 (67.4)	46 (67.6)		22 (47.8)	44 (64.7)	
*BRCS*						
Total scores	11.5 ± 3.9	11.8 ± 3.8	3.92	13.2 ± 3.3	10.8 ± 4.2	−3.18 **
Resilience levels (tertiles)						
Low (4–10)	18 (39.1)	25 (36.8)	0.12	10 (21.7)	30 (44.1)	6.08 *
Medium (11–14)	19 (41.3)	28 (41.2)		19 (41.3)	21 (30.9)	
High (≥15)	9 (19.6)	15 (22.0)		17 (37.0)	17 (25.0)	

* *p* < 0.05, ** *p* < 0.01; ^a^ T0: January–December 2018 (baseline interview), T1: January–May 2020; ^b^ FG: follow-up group; CG: control group; BSRS-5: Brief Symptom Rating Scale; BRCS: Brief Resilient Coping Scale.

## Data Availability

Data will be available from the corresponding author on reasonable request.

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
