# Peer review of "The Influence of Research Follow-Up during COVID-19 Pandemic on Mental Distress and Resilience: A Multicenter Cohort Study of Treatment-Resistant Depression"

_ijerph, 2022, doi:10.3390/ijerph19063738_

Round 1
Reviewer 1 Report
The work has an original character because it was made in wide geography and time. It is a study that can contribute to the literature in terms of the impact of Covid-19 on people, but I have listed some issues that need to be corrected and completed under the relevant headings below. I recommend that you do your future studies by using more samples and establishing a model.
Abstract
- The section up to Methods in the abstract does not fully explain the purpose of the study or is not understood. Please review.
- Briefly describe what you are working on in the follow-up group (in regular research engagement) in summary.
Introduction
- In the introduction, the study should have a research question.
- It is not written on which theory the study is based on. And that theory should be explained separately.
- The hypotheses of the study that should be underpinned by the literature have not been written.
- The literature is not written enough. In particular, the literature to support the hypotheses to be written should be compiled with current data.
Method, Results, and Discussion
- Under the heading Measures, the variables used under the heading Personal demographics variables should be written.
- Basic analyzes were used. However, couldn't analyzes be done with more advanced modeling?
- The hypotheses to be written should be explained in the discussion section by comparing them with the current literature.
- The research question to be written should be evaluated in line with the results of the hypotheses and current literature at the end of the discussion part. If possible, the discussion part should be written with two subtitles as hypotheses and research question.
- Limitations paragraph should be separated from the discussion and written under a separate heading.
- Conclusions should be further detailed with Implications.
- References are insufficient. There are many studies on the psychological effects of Covid-19. I listed some below for you to use.
Dong, Z.; Ma, J.; Hao, Y.; Shen, X.; Liu, F.; Gao, Y.; Zhang, L. The social psychological impact of the COVID-19 pandemic on medical staff in China: A cross-sectional study. Eur. Psychiatry 2020, 63, E65.
Flanagan, E.W.; Beyl, R.A.; Fearnbach, S.N.; Altazan, A.D.; Martin, C.K.; Redman, L.M. The impact of COVID-19 stay-at-home orders on health behaviors in adults. Obesity 2021, 29, 438–445.
Koçak, O.; Koçak, Ö.E.; Younis, M.Z. The Psychological Consequences of COVID-19 Fear and the Moderator Effects of Individuals’ Underlying Illness and Witnessing Infected Friends and Family. Int. J. Environ. Res. Public Health 2021, 18, 1836. https://doi.org/10.3390/ijerph18041836
Author Response
Thank you very much for giving us a chance to revise. We appreciate your constructive opinions that brought insights for us and facilitated our manuscript revision. Attached please find our responses toward each comment underlie or in italics.

Reviewer 2 Report
It is a good attempt, given the idea is still so new. So, cannot hold authors responsible for adequate literature review. However, the authors may want to add some background on papers done on similar issues from a psychological or clinical perspective (with respect to the methodology employed, what was the result).
The manuscript requires significant improvement in writing. I suggest professional proofreading.
I am not sure whether one could draw inference other than to state observation from the methodology used by authors. It is simply descriptive. Significant work is required for improving the methodology to suit the question being asked. I do understand that regression analysis and such may not be possible due to data limitation, however, as in most clinical trial studies, the non-parametric methodology may be suitable for this study.
This approach differs from classical models in that it does not rely on strong assumptions regarding the shape of the relationship between the variables. The bootstrap method may be more apt for the study.
For more resources see (British Psychological Society): https://thepsychologist.bps.org.uk/volume-22/edition-5/methods-giving-your-data-bootstrap
Author Response
Thank you very much for giving us a chance to revise. We appreciate your constructive opinions that brought insights for us in manuscript writing. Attached please find our responses toward each comment underlie or in italics.

Reviewer 3 Report
This is well written manuscript on current interesting topic. However, some points should be addressed for improving and publication of this study. My comments are as belows:
[introduction]
#1. Add previous study regarding resilience in patients with mental disorder, especially MDD, to stress the rationale of this study.
[methods]
#2. This study is not pre-defined study. During COVID-19 outbreak in 2020, the research team had a fortunate opportunity to observe the TRD cohort in terms of the levels of resilience and psychological distress before and during the pandemic. This seeme to be different from the purpose of original study. Therefore you should address issue on new IRB approval. And add this as limitation in the discussion section.
#3. Specify standardized procedure of the study interviews (for example, process and method of development of standardized procedure, total time, specific procedure, traing procedure, background of researcher, and so on). And it would be better to provide any figure to visulalize total procedure for readers.
#4. Check title of table 2. Use abbreviation within the table 2 and 3. (The 9-item Concise Mental Health Check-list =>CMHC-9, the others are same.)
#5. Figure legends.
(Figure 1. The fluctuations of mental distress across COVID-19 between the two groups; Follow-up 201 group: M = 12.4, SD = 6.6 to M = 9.8, SD = 6.7; Control group: M = 11.8, SD = 5.9 to M = 11.1, SD = 7.2)
"Follow-up 201 group: M = 12.4, SD = 6.6 to M = 9.8, SD = 6.7; Control group: M = 11.8, SD = 5.9 to M = 11.1, SD = 7.2" is not appropriate. Delete this.
[discussion]
#6. Thess sentences seem to fit in introduction. Move to intro section.
"Patients with mental disorders are probably at higher risk of relapse due to fear re-225 lated to the COVID-19 outbreak [28]. Current research on resilience and mental distress 226 under the COVID-19 pandemic has been scarce, with only a few cross-sectional studies 227 concerning young adults [8] and healthcare workers [29]. However, resilience was a pro-228 tective factor to stress during and after the pandemic [12, 14]. A study in SARS survivors 229 after 6, 12, 18 months of follow-up identified that those with higher levels of resilience and 230 recovery experienced less SARS-related anxiety and more significant social support [30]."
#7. Provide proper reference in this sentence.
"A possible reason for this improvement may remain due to the Hawthorne effect."
#8. Contrast and compare your results using previous relevant studies for psychological benefit or Hawthorne effect of regular interviews.
#9. Use subtitle in this section for readablility such as 1) Summary of results in this study 2) interpretation of this results 3) clinical implication 4) limitation and future study direciton
#10. You talked only two points on limitation of this study. Add limitations of this study using relevant reference.
#11. Add clinical implication of this study.
#12. Provide the strength of this study
#13. Provide specific direction of future research using relevant reference.
#14. conclusion
Specific conclusions should be based on overall interpretation of findings and limitation. I think re-write conclusion considering this.
Author Response

(The authors gave the same response as above.)

Round 2
Reviewer 1 Report
The work is now better than before. Your analyzes and literature are advanced. Some English corrections may be required. Congratulations.
Author Response
Thank you very much for your feedback and approval for our manuscript.

Reviewer 2 Report
Since authors are not in agreement to improving their manuscript, I have no other option but to recommend rejection in its current format.
Author Response
Dear reviewer, we apologize for the misunderstanding. We appreciate your advice and have added some information in the sections of Introduction, Discussion, and Conclusion. Please refer to the updated revisions in the attached file for your reference. We appreciate your further decision of acceptance of our revised manuscript.

Reviewer 3 Report
Some points still should be addressed. My comments are belows:
#1. You should cite not general review but direct relevant reference for resilience in patients with MDD or depression.
#2. You hypothesized that the research follow-ups through the outbreak would enhance the psychological conditions of patients with TRD due to provision of perceived social support. However, your study was approved before COVID-19 outbreak in 2020. Therefore, your protocol for IRB was not consider COVID-19 outbreak. You should address this point properly.
#3. This points was not addressed well.
"Specify standardized procedure of the study interviews (for example, process and method of development of standardized procedure, total time, specific procedure, training procedure, background of researcher, and so on). And it would be better to provide any figure to visulalize total procedure for readers."
#11. #13.
"Add clinical implication of this study"
"Provide specific direction of future research using relevant reference."
These points was not addressed well. And you should write this specific point not in conclusion but in discussion section.
#14. These points was not addressed well.
Specific conclusions should be clear and based on overall interpretation of findings and limitation. You avoid subjective and ambiguous expression not based on your findings. Some sentences is proper not in conclusion but specific discussion. I still think you should change conclusion considering these points.
Author Response
Thank you very much for giving us a chance to have the second round of revisions. We appreciate your constructive opinions that brought insights for us and facilitated our manuscript revision. Attached please find detailed contents for your reference.
